# A Dynamic Mixup Approach Towards Improved Robustness of Classifiers

## Abstract

The robustness of image classifiers has been an extensive area of growing research. Current methods typically rely on data augmentation techniques to simulate distribution shifts based on image corruptions. These techniques mainly consider linearity to generate synthetic samples. However, corruptions that occur in the real world are more complex and unlikely to follow a linear drift. We introduce an adaptation of the *mixup* approach, Dynamic Mixup, as our data processing technique. Dynamic Mixup uses a simple mixing strategy to combine augmented versions considering the non-linearity that exists between them. Training on these samples encourages learning representations robust to new or unseen distortions. Our experimental findings reveal that Dynamic Mixup outperforms the previous methods with improved robustness in image and object classification tasks.

## 1 Introduction

Image classifiers are trained on large datasets of labeled images. However, these datasets may not be representative of the real world, where the distribution of images can vary significantly. We encounter naturally diverse, realistic, and photo-realistic images in practical scenarios. While these images are easily recognizable to human eyes, they frequently pose a challenge for computer vision tasks (Tokozume et al.; Liu et al.; PaK & Kim; Minaee et al.) due to the distribution shifts they introduce. As a result, the performance of Deep Neural Networks (DNNs) degrades since the test data characteristics differ from those of the training data (Jiashuo et al.).

Data shift or distribution shift refers to the change in the distribution of images between the training and test sets. This can be caused by a variety of factors, such as changes in lighting, camera angle, or scene context. Data shift is a major challenge for image classifiers, as it can lead to significant performance degradation. For example, an image classifier trained on images of cars in sunny weather may perform poorly on images of cars in rainy weather. Even a minor corruption in data distribution can potentially topple the prediction of modern classifiers. Previous studies reveal that one of the major reasons affecting the robustness of DNNs on image classification is data distribution shift (Jiashuo et al.) caused by adversarial noise (Chakraborty et al.) and common image corruptions (Hendrycks & Dietterich). Instances where a potential failure due to distortions is visible indicate unreliable predictions (Hendrycks & Dietterich; Dodge & Karam; Geirhos et al.). Therefore, a crucial need exists for a learning strategy to adequately capture intricate visual details within corrupted images, preventing significant distortion from notably impacting prediction outcomes.

Among the existing methods (Mendonca et al.; Croce et al.; DeVries & Taylor; Goldblum et al.) that address robustness in DNNs, data augmentation is widely adopted either at its core (Hein & Andriushchenko), in conjunction with, or as a primary technique (Rebuffi et al.; Hendrycks et al.). Data augmentation uses transformation operations and aims to create synthetic samples that closely resemble data in practical scenarios. It is useful in situations where there is a scarcity of data or limited availability of labeled data. A common observation is that most of the augmentation methods emphasize linearity while producing training samples to replicate distribution shifts. Also, these samples are mostly generated using common corruptions or adversarial perturbations. Other studies (Chun et al.) state that many techniques focus on improving clean accuracy but robustness is compromised. Specifically, some previous works imply a fundamental trade-off between accuracy and robustness (Tsipras et al.; Zhang et al.).

It is important to note that data shifts can occur even if the training and test sets are drawn from the same underlying distribution. For example, the distribution of images can shift over time due to changes in technology. When developing image classifiers for real-world applications, it is important to consider the potential for data shift and to take steps to improve the robustness of the classifier. Naturally occurring distribution shifts can range from common corruptions, lower-level distortions, such as motion blur and illumination changes, to semantic ones, like object occlusion (Kar et al.). We argue that it is intuitively unlikely for the distortions to follow a linear drift. Therefore, linear interpolation is not sufficient to capture desired features in simulated corruptions that are more likely to appear in practice. Consequently, the estimation of robustness in classification tasks should not be confined to common corruptions and perturbations alone.

In this work, we highlight a fundamental behavior of transfer characteristics occurring during distribution shifts in data as shown in Figure 1. We approach the problem of robustness in image classification by focusing on the linear and non-linear behavior observed during shifts in data. We propose 'Dynamic Mixup', an adaptation of the mixup (Zhang et al.) approach, as our data processing technique. Dynamic Mixup dynamically mixes augmented image samples, considering the non-linearity that exists between them. We show that our mixing strategy enables the model to capture intricate details and learn more robust representations that are less sensitive to distribution shifts in the natural environment. We build upon the seminal state-of-the-art work presented in prior studies, employing a similar suite of experiments to rigorously assess the efficacy of our proposed approach in improving robustness and uncertainty estimates. Our experimental findings show significant improvement in the robustness of classification tasks over a number of previous data augmentation techniques. Dynamic Mixup reduces the corruption robustness error of standard training procedures from 29% to 10% on CIFAR-10 and 55.6% to 35.5% on CIFAR-100. We also achieve improved results on ImageNet where the corruption error and perturbation instability decrease from 80.6% to 64% and 57.2% to 33.3%, respectively. Additionally, on ImageNet-3DCC, the corruption robustness error on the object classification task significantly reduces from 85.6% to 66.1%.

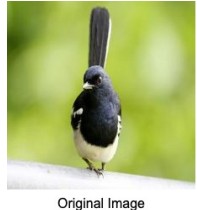 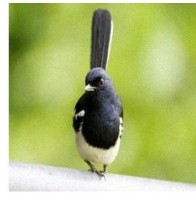

(a) A random Gaussian noise applied to a bird's image.

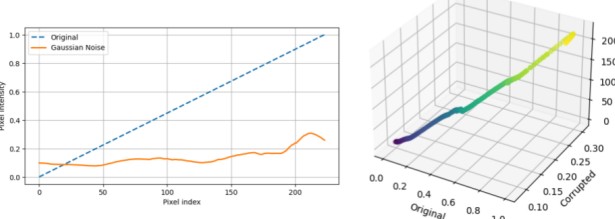

(b) 2D-plot (left) showing the relationship between the pixel intensities of the original image and image with Gaussian noise based on the MSE values. 3D plot (right) for better interpretation of this drift across pixel intensities.

Figure 1: We show the nature of drift in original data data under common corruption. In the case of naturally occurring noises (external) and distortions (signal-induced), we expect substantial deviation that is unlikely to follow a linear drift and more likely to resemble a *spline curve*. Thus, interpolation should be based on both linear and non-linear characteristics to enforce learning of robust representations.

## 2 RELATED WORK

**Robustness of DNNs.** Most research on robustness to distribution shift focuses on synthetic image perturbations, such as noise, simulated weather artifacts, and adversarial examples. However, it is unclear how well robustness to synthetic distribution shift translates to robustness to distribution shift arising in real data (Chakraborty et al.). Moreover, the perfectly fitted DNNs tend to memorize specific training distortions for better test performance (Ovadia et al., Geirhos et al.). Solving this generalization problem is crucial for robust inference. A study on the robustness of transformers for image classification Bhojanapalli et al. suggests that more pre-training data improves performance on out-of-distribution data. Meanwhile, (Hendrycks & Dietterich) establishes benchmarks for measuring the generalization of classifiers to unseen corruptions. For adversarial perturbations, a similar benchmark (Kaufmann et al.) exists. Thus, robustness to distribution shift is critical for the reliability of machine learning systems (Gilmer et al.).

**Data augmentation.** Data augmentation is a powerful technique to improve generalization performance in applications such as image classification tasks (Wu et al.). Recent studies have used diverse tactics to apply data augmentation other than commonly followed random left-right flipping and cropping (He et al.). Cutout uses random occlusion techniques (DeVries & Taylor), CutMix (Yun et al.) replaces a portion of an image with a a portion of different image, and Mixup (Zhang et al.) uses information from two different images to mix them. Meanwhile, AutoAugment (Cubuk et al.) and Deep AutoAugment (Zheng et al.) automatically search for data augmentation policies via reinforcement learning that involves a policy search cost. On the other hand, AugMix (Hendrycks et al.) which produced state-of-the-art results, is comparatively simpler and uses a convex chain of transformation sequences with varying depths of operations to produce an augmented version that is later mixed with the original image. The process repeats twice to produce mixed images. In our work, we employ a dynamic selection of augmented operations that are composed to form an augmented sample of the original image, which is then combined with another augmented version of the same image.

**Calibration for robustness under shift.** Calibration holds significance in safety-critical applications as it plays a pivotal role in quantifying uncertainty and informing decision-making processes (Pampari & Ermon). A model is said to be robust if it is well-calibrated. Calibration indicates the confidence of prediction probabilities in a machine learning model. In other words, an estimate of how well we can trust the model's prediction in practical situations. (Guo et al.) assessed a number of calibration methods, the impact of hyperparameters on the calibration of a network, and suggested metrics for evaluation. (Gong et al.) discuss covariate shifts and leverage multiple calibration domains for domain generalization to calibrate classifiers without using any data from the target domain. (Krishnan & Tickoo) introduce a loss function to improve calibration in uncertainties. Another method proposed in (Wenzel et al.) uses the ensembling of hyperparameters to boost robustness and uncertainty while maintaining generalization.

## 3 DYNAMIC MIXUP

Dynamic Mixup is a data augmentation technique that improves the robustness of classification tasks in unforeseen environments. It utilizes a diverse set of transformations and a mixing strategy to combine the augmented versions. The mixing strategy enables the generated samples to capture both linear and non-linear components of the data shift. On-the-fly application of random transformations and the mixing strategy encourages the model to learn the intricacies between different data distributions. We then use the Jensen-Shannon divergence (Menéndez et al.) as a consistency loss to enforce consistency across the mixed versions of the original image. In this way, the augmented versions capture the features well with diverse transformations and yet do not drift too far from the original image. The learned features enable robust predictions and generalize well to unseen corruptions and other naturally induced distortions.

As training against specific distortions causes the classifiers to memorize them, mixing allows us to create a diverse set of training samples. The general idea behind mixing is to introduce heterogeneity in the augmented samples, which drives the model towards enhanced robustness. Prior methods have used data augmentation in a linear or convex chain of transformation sequences. Directly composing augmentations in a cascade of operations may cause the image to quickly degrade and

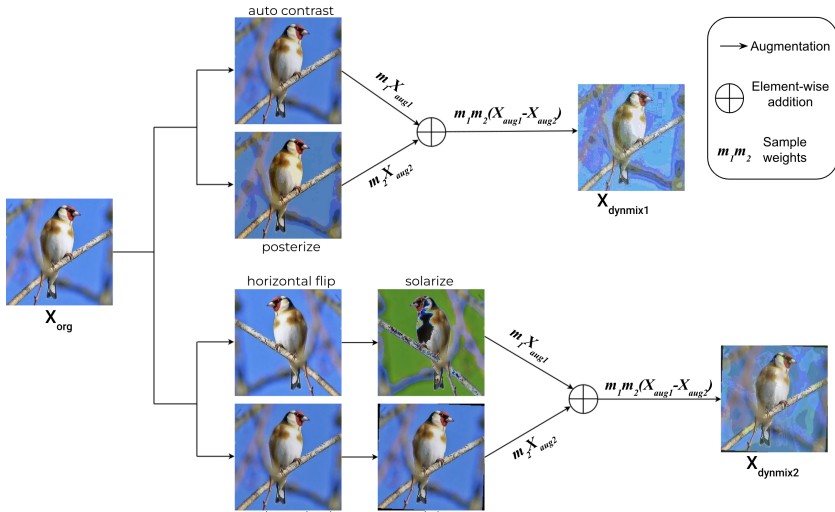

Figure 2: A high-level overview of Dynamic Mixup augmentation strategy. Transformation operations such as Auto Contrast and Posterize are randomly sampled and mixed using mixing coefficients for sample weights considering both linear and non-linear behavior of data shift. Using this scheme, two mixed images are produced for a given input image, and the model is encouraged to learn the intricate transfer characteristics crucial for improved robustness.

drift off the data manifold (Hendrycks et al.). Furthermore, mixing disjunct images may also cause the composition to diverge from the original image (see Appendix C) and thus, misclassification due to the lucid presence of more than one category. We provide a solid understanding of our algorithm in the pseudocode below.

---

**Algorithm** Pseudocode for Dynamic Mixup

---

1: **Input:** Model $\hat{p}$, Classification Loss $\mathcal{L}$, Image $X_{org}$, Transformations $T = \{rotate,...,equalize\}$
2: **function** DynamicMixup($X_{org}$, k=2, $\alpha$=1)
3:    $X_{aug} \leftarrow 0$
4:    **for** i=1 to k **do**
5:       Randomly sample transformations $t_1, t_2,... \sim T$
6:       Compose transformations with depth $t_{12} = t_2 \circ t_1$
7:       Sample randomly from one of these transformations chain$\sim\{t_1, t_{12}\}$
8:       $X_{aug}$[i] += chain($X_{org}$)
9:    **end for**
10:   Sample weight $m \sim$ Beta($\alpha, \alpha$)
11:   Interpolate $X_{dynmix} = m_1 X_{aug1} + m_2 X_{aug2} + m_1 m_2 (X_{aug1} - X_{aug2})$
12: **end function**
13: $X_{dynmix1} = $ DynamicMixup($X_{org}$)              $\triangleright X_{dynmix1}$ is stochastically generated
14: $X_{dynmix2} = $ DynamicMixup($X_{org}$)                  $\triangleright X_{dynmix1} \neq X_{dynmix2}$
15: **Output:** $\mathcal{L}(\hat{p}(y|X_{org}), y) + \lambda JS(\hat{p}(y|X_{org}); \hat{p}(y|X_{dynmix1}); \hat{p}(y|X_{dynmix2}))$

---

**Augmentation.** We mix the results of transformation compositions to produce an augmented (mixed) version of the original image. A pair of augmented versions are generated as training samples. On CIFAR-10 and CIFAR-100, we use a set of 11 operations, each operation is visualized. In addition to this, we apply a downscale operation while processing ImageNet data (Figure 5, 6). To fairly evaluate our method on the robustness benchmark, we refrain from using operations that coincide with ImageNet-C corruptions. Particularly, we avoid color, brightness, contrast, sharpness, and any cutout operations in addition to noises like shot, fog, blur, etc. Doing so ensures that the corruptions used by ImageNet-C are only encountered during inference. Operations such as rotate, translate, and shear can vary in severity magnitudes, for instance, 10 or -2 degrees for rotate oper-

ation and so on. The augmentation process bifurcates into k=2 augmentation chains where at most two randomly selected transformations can be composed.

**Mixing.** We combine the resulting images through mixing. The essence of our mixing strategy can be realized in the following to interpolate between the existing data points:

$$m_1 X_{aug1} + m_2 X_{aug2} + m_1 m_2 (X_{aug1} - X_{aug2})$$

Here, $m_1$ and $m_2$ are the sample weights for mixing. We draw these weights randomly from a Beta($\alpha, \alpha$), where $\alpha$ is the mixing coefficient and Beta is used as a continuously updated probability distribution. $X_{aug1}$ and $X_{aug2}$ are a composition of transformation operations. While $m_1 X_{aug1} + m_2 X_{aug2}$ is responsible for linear interpolation, the residual term $m_1 m_2 (X_{aug1} - X_{aug2})$ captures non-linearity between the data points. The non-linearity introduced by the difference term can be helpful in data augmentation as it helps to create new data points that are more representative of the real world where data is often not linearly distributed. There are often non-linear relationships between variables. By adding non-linearity to the interpolation, we can create new data points that better reflect these non-linear relationships. The final mixed image is thus endowed with randomness from different operations, their severity level, and the mixing coefficients (weights) for the interpolation of the augmented samples.

**Consistency Loss.** It is important to strike a balance between the amount of linearity and non-linearity for desirable intermediate characteristics that we require in the mixed images to be used in training. For smoother network responses, it is crucial to use a loss that enforces consistency across the cognate sample embeddings ($X_{org}, X_{dynmix1}, X_{dynmix2}$) as they are supposed to drift not far enough to preserve the semantics of the image. To achieve this, we incorporate a consistency loss (Laine & Aila) with our augmentation strategy. The consistency loss penalizes the network if it predicts the same unlabeled data inconsistently given different augmentations. Hence, we first aim to minimize the Jensen-Shannon divergence among the posterior distributions of the original image and its augmented versions:

$$p_{org} = \hat{p}(y|X_{org}), p_{dynmix1} = \hat{p}(y|X_{dynmix1}), p_{dynmix2} = \hat{p}(y|X_{dynmix2}).$$

The original loss $\mathcal{L}$ now becomes:

$$\mathcal{L}(p_{org}, y) + \lambda JS(p_{org}; p_{dynmix1}; p_{dynmix2})$$

During each forward pass, the model receives the original image ($X_{org}$)along with two augmented versions, namely $X_{dynmix1}$ and $X_{dynmix2}$. The intent is to encourage the model to produce consistent predictions across various augmentations of the same input data, prompting enhanced learning of representations. To achieve this, the JS divergence is incorporated into the entropy loss, weighted by the lambda hyperparameter. Thus, for simplicity, the Jensen-Shannon Consistency Loss encourages the predicted probability distribution to be close to the mean of the predicted and true probability distributions. In other words, the model will be less likely to overfit the noisy data and will be more likely to generalize well to unseen data. We explain this loss in detail in Appendix A.

## 4 EXPERIMENTS

**Datasets.** We use two CIFAR (Krizhevsky) datasets and ImageNet (Deng et al.) in our training. Both CIFAR-10 (10 classes) and CIFAR-100 (100 classes) contain small 32x32x3 natural color images where 50,000 are used for training and 10,000 as testing images. ImageNet has 1000 classes with nearly 1.2 million large-scale color images. For evaluation, we use benchmarks designed specifically for the assessment of the model's robustness to common corruptions (CIFAR-10-C, CIFAR-100-C, ImageNet-C) and perturbations (ImageNet-P) as introduced by (Hendrycks & Dietterich). These datasets are created by corrupting the original CIFAR and ImageNet test sets. Each dataset encompasses 15 types of noise, blur, weather, and digital corruptions varying across 5 severity levels. We do not introduce these corruptions during training to avoid overlapping and gauge the network's behavior towards these shifts. While CIFAR-10-C and CIFAR-100-C are image datasets, Imagenet-P contains videos as a sequence of frames. Furthermore, we use Imagenet-3DCC (Kar et al.), available as a benchmark to measure the resilience of neural networks on more complex corruptions (near focus, far focus, fog 3D) based on geometric transformations and semantic corruptions that closely resemble realistic data shifts.

**Metrics for robustness.** Generally, we use the Clean Error on the clean or uncorrupted test data. The next step is to test the classifier on each corruption type. For corruption of type c at each level of severity $s$, where $(1 \leq s \leq 5)$, the error rate is $E_{c,s}$. The unnormalized corruption error $uCE_c = \sum_{s=1}^{5} E_{c,s}$ is calculated by taking the average error across severities for a given corruption type. We average these errors for 15 corruptions in the case of CIFAR-10-C and CIFAR-100-C. On ImageNet, we follow the convention and normalize the corruption error by AlexNet's (Krizhevsky et al.) corruption error $CE_c = \sum_{s=1}^{5} E_{c,s}^{AlexNet}$. Thus, the model's corruption robustness can be summarized by averaging the 15 Corruption Error values $CE_{GaussianNoise}, CE_{ShotNoise}, ..., CE_{Pixelate}, CE_{JPEG}$. This results in the *mean Corruption Error (mCE)*. The mCE on Imagenet-3DCC is calculated likewise.

We estimate our model's uncertainty by measuring its calibration error or miscalibration on corrupted data. A model is said to be 'calibrated' when it is capable of producing reliable predictions in uncertain environments. Thus, if we are to assess the samples that were estimated to be positive with a probability of 0.85, we would expect that 85% of them are positive. Mathematically, for a given probability value $p$, a class prediction is correct 100*$p$ percent of the time. We use the RMS Calibration Error to assess uncertainty. To interpret this, let $C$ be the classifier's confidence that its prediction $\hat{Y}$ is correct. Thus, the squared difference or RMS Calibration Error between the accuracy at a given confidence level and the actual confidence is given by $\sqrt{E_C[(P(Y) = \hat{Y} \mid C = c) - c]^2}$.

For estimating robustness to perturbations, we measure the "jaggedness" or how well the video frame predictions of a sequence match. Thus, we calculate the flip probability as we do not want the network to give inconsistent predictions between frames of the video as the brightness increases. For simplicity, if in a video sequence, any two adjacent frames or frames with slightly different brightness have a mismatched prediction, we term it as "flipped". To calculate the *mean Flip Probability (mFP)* we take the average error across 10 different perturbation types. Similar to ImageNet-C, we normalize with AlexNet's flip probabilities and compute the *mean Flip Rate (mFR)*.

Table 1: Average classification error(%) across several architectures. The corruption robustness of Dynamic Mixup (Ours) on CIFAR-10-C improves over previous methods.

|  | Standard | Mixup | CutMix | AA* | Adv. Training | AugMix | DynMix |
|---|---|---|---|---|---|---|---|
| AllConvNet | 30.8 | 24.6 | 31.3 | 29.2 | 28.1 | 15.0 | **12.8** |
| DenseNet | 30.7 | 24.6 | 33.5 | 26.6 | 27.6 | 12.7 | **10.1** |
| WideResNet | 26.9 | 22.3 | 27.1 | 23.9 | 26.2 | 11.2 | **8.9** |
| ResNeXt | 27.5 | 22.6 | 29.5 | 24.2 | 27.0 | 10.9 | **8.6** |
| Mean | 29.0 | 23.5 | 30.3 | 26.0 | 27.2 | 12.5 | **10.1** |

Table 2: Average classification error(%) on CIFAR-100-C across several architectures. Our method performs better than other methods.

|  | Standard | Mixup | CutMix | AA* | Adv. Training | AugMix | DynMix |
|---|---|---|---|---|---|---|---|
| AllConvNet | 56.4 | 53.4 | 56.0 | 55.1 | 56.0 | 42.7 | **40.2** |
| DenseNet | 59.3 | 55.4 | 59.2 | 53.9 | 55.2 | 39.6 | **36.7** |
| WideResNet | 53.3 | 50.4 | 52.9 | 49.6 | 55.1 | 35.9 | **32.8** |
| ResNeXt | 53.4 | 51.4 | 54.1 | 51.3 | 54.4 | 34.9 | **31.7** |
| Mean | 55.6 | 52.6 | 55.5 | 52.5 | 55.2 | 38.3 | **35.5** |

## 4.1 CIFAR-10 AND CIFAR-100

**Training.** We experiment across various methods and architectures to demonstrate the effectiveness of Dynamic Mixup in improving the robustness of classification tasks on CIFAR-10 and CIFAR-100. We train on ResNeXt-29 (32d × 4) (Xie et al.), Wide ResNet (Zagoruyko & Komodakis), DenseNet-BC (k = 12, d = 100) (Huang et al.), and All Convolutional Network (Springenberg et al.). The initial learning rate is set to 0.1 for all networks which gradually decays following the cosine annealing

schedule. We apply random cropping and resize all input images before applying any augmentations. To ensure consistency, we preserve parameters while experimenting across CIFAR-10 and CIFAR-100. While All Convolutional Networks and Wide ResNet train for 100 epochs, the bigger networks DenseNet and ResNeXt take 200 epochs for convergence. As for stochastic gradient descent, we use Nesterov momentum for optimization. The weight decay that we use for mixup is 0.0001 and 0.0005 for other methods.

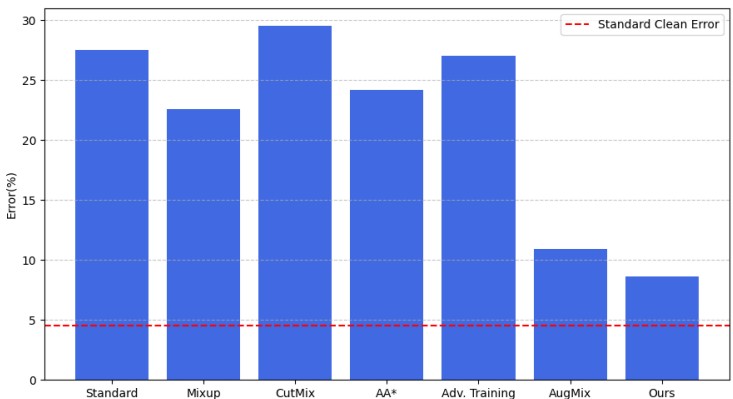

Figure 3: We show error rates of various methods on CIFAR-10-C using a ResNeXt backbone. Dynamic Mixup minimizes the error rate of previous methods and progresses towards the clean error rate.

**Results.** We observe that Dynamic Mixup shows improved performance over previous methods that we reproduce in our experiments. Our dynamic mixing strategy combined with Jensen-Shannon loss results in increased robustness. With Dynamic Mixup, the absolute corruption error of the Standard augmentation method is reduced by 19% on CIFAR-10-C using ResNeXt. Notably, the performance of ResNeXt is superior to other networks as it proceeds towards the clean error rate (Figure 3). We show the details of all methods in Table 1. Our method outperforms other data augmentation techniques. In addition to this, the performance improves consistently across different neural network architectures, and on CIFAR-100-C since it does not require further fine-tuning, results are provided in Table 2.

## 4.2   IMAGENET

**Training.** We show the viability of Dynamic Mixup by comparing it with several data augmentation methods that work well on large-scale images, specifically on the ImageNet dataset. We perform our experiments with AutoAugment*, Random AutoAugment*, Stylized ImageNet (SIN) (Geirhos et al.), and AugMix. As we do not use AutoAugment augmentations that overlap with ImageNet-C corruptions, we denote it as AutoAugment*(or AA*) in the results. Random AutoAugment* is denoted similarly, and applies a random augmentation policy using AutoAugment operations. SIN uses a style-transfer approach to avoid texture bias and retain shape semantics, models are trained on original as well as ImageNet images with style transfer applied. Since AutoAugment utilizes policy search, it has higher computation than techniques like Random AutoAugment* and AugMix. After a few iterations of tuning, we find per previous results in (Hendrycks et al.), that the mCE reduces by 0.6% when using 0.5 as content and style loss coefficients instead of 0 and 1.

We pre-process all input images with standard random cropping and resize them. Next, we train models on these methods from scratch using ResNet-50 using the training scheme followed by (Goyal et al.) where the learning rate is linearly scaled with the batch size and uses a warm-up for the first 5 epochs. Meanwhile, AutoAugment, AugMix, and Dynamic Mixup train for 180 epochs. We apply standard random cropping and horizontal flipping to all input images while reproducing the results for other methods.

**Results.** We find that the robustness in performance is steady even beyond the aggregated measures across every individual corruption and severity level even on ImageNet-C as well. Using Dynamic

Table 3: Clean Error, Corruption Error (CE), and mCE values for various methods on ImageNet-C. The mCE value is computed by averaging across all 15 CE values.

| Network | Clean | Noise | | | Blur | | | | Weather | | | | Digital | | | | mCE |
| --- | --- | --- | --- | --- | --- | --- | --- | --- | --- | --- | --- | --- | --- | --- | --- | --- | --- |
| | | Gaussian | Shot | Impulse | Defocus | Glass | Motion | Zoom | Snow | Frost | Fog | Bright | Contrast | Elastic | Pixel | JPEG | |
| Standard | 23.9 | 79 | 80 | 82 | 82 | 90 | 84 | 80 | 86 | 81 | 75 | 65 | 79 | 91 | 77 | 80 | 80.6 |
| AA* | 22.8 | 69 | 68 | 72 | 77 | 83 | 80 | 81 | 79 | 75 | 64 | 56 | 70 | 88 | **57** | 71 | 72.7 |
| Random AA* | 23.6 | 70 | 71 | 72 | 80 | 86 | 82 | 81 | 81 | 77 | 72 | 61 | 75 | 88 | 73 | 72 | 76.1 |
| SIN | 27.2 | 69 | 70 | 70 | 77 | 84 | 76 | 82 | 74 | 75 | 69 | 65 | 69 | 80 | 64 | 77 | 73.3 |
| AugMix | 22.4 | 65 | 66 | 67 | 70 | 80 | 66 | 66 | 75 | 72 | 67 | 58 | 58 | 79 | 69 | 69 | 68.4 |
| **DynMix** | **18.3** | **60** | **61** | **63** | **66** | **75** | **61** | **63** | **71** | **68** | **63** | **52** | **51** | **78** | 69 | **65** | **64.4** |

Table 4: The mean flipping rate across all 10 perturbation types on ImageNet-P, perturbation stability using Dynamic Mixup improves by approximately 24%.

| Network | Clean | Gaussian | Shot | Motion | Zoom | Snow | Bright | Translate | Rotate | Tilt | Scale | mFR |
| --- | --- | --- | --- | --- | --- | --- | --- | --- | --- | --- | --- | --- |
| Standard | 23.9 | 57 | 55 | 62 | 65 | 66 | 65 | 43 | 53 | 57 | 49 | 57.2 |
| AA* | 22.8 | 50 | 45 | 57 | 68 | 63 | 53 | 40 | 44 | 50 | 46 | 51.7 |
| Random AA* | 23.6 | 53 | 46 | 53 | 63 | 59 | 57 | 42 | 48 | 54 | 47 | 52.2 |
| SIN | 27.2 | 53 | 50 | 57 | 72 | 51 | 62 | 43 | 53 | 57 | 53 | 55 |
| AugMix | 22.4 | 46 | 41 | 30 | 47 | 38 | 46 | 25 | 32 | 35 | 33 | 37.4 |
| **DynMix** | **18.3** | **41** | **37** | **28** | **41** | **33** | **41** | **23** | **28** | **31** | **30** | **33.3** |

Mixup the corruption robustness error improves as we achieve an mCE of 64.4% (Table 3). This is much lower than the baseline mCE of 82.1%. As shown in Table 4, our method exceeds previous results on ImageNet-P with an mFR of 33.3% compared to the standard 58.7% mFR. We also show the mCE for object classification on Imagenet-3DCC in Figure 4 by plotting in parallel the mCE of ImageNet-2DCC (ImageNet-C) and comparing our results with AugMix. ImageNet-3DCC contains 3D common corruptions and Dynamic Mixup achieves a comparable error rate when exposed to such complex corruptions. Furthermore, we provide results for uncertainty estimates of the classifier on ImageNet corrupted data in Figure 9 (Appendix D). It is noteworthy that, when subjected to significant data shifts, Dynamic Mixup exhibits a consistent RMS calibration error. Specifically, despite an increase in classification error, calibration remains relatively stable.

Table 5: Ablation results of Dynamic Mixup on CIFAR-10-C and CIFAR-100-C.

| | CIFAR-10-C error | CIFAR-100-C error |
| --- | --- | --- |
| Standard | 26.9 | 53.3 |
| No JS Loss (AugMix) | 13.1 | 39.8 |
| No JS Loss (DynMix) | 10.4 | 35.6 |
| Only linear component (DynMix) | 13.3 | 39.2 |
| JSD with single sample (AugMix) | 12.6 | 37.3 |
| JSD with single sample (DynMix) | 9.7 | 34.1 |

### 4.3 ABLATIONS.

We perform ablation experiments to assess the changes in the performance of Dynamic Mixup under various test cases using Wide ResNet. Results are provided in Table 5. It is noticeable that removing the Jensen-Shannon Loss leads to less control on the consistency as the error rate increases from 8.9% to 10.4% on CIFAR-10-C and 32.4% to 39.8% on CIFAR-100-C. We also observe the impact of considering only the linear component of drift ($m_1 X_{aug1} + m_2 X_{aug2}$) during mixing as the error increases to 13.3% on CIFAR-10-C and 39.2% on CIFAR-100-C. Next, we use a single augmented

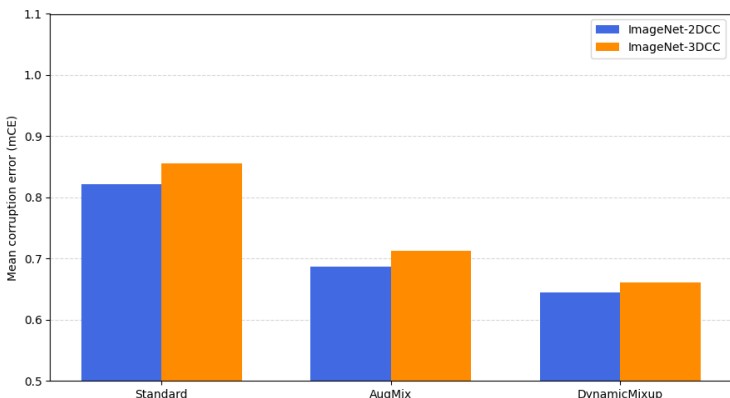

Figure 4: A comparison of robustness on ImageNet-3DCC and ImageNet-2DCC (ImageNet-C). We notice that Dynamic Mixup gains are apparent on a challenging dataset with complex corruptions.

version, instead of a pair to compute the error rate. Although the error increases to 9.7%, it remains on par with the average classification error using Dynamic Mixup.

## 5 CONCLUSION

We conducted our study based on a subtle yet non-trivial aspect of distribution shift that required attention and observed nuances of including it in formulating our strategy for Dynamic Mixup. We use a simple data augmentation technique with little or no overhead of any policy search as it uses a dynamic mixing strategy to apply random augmentations. It combines the generated augmented versions to produce a mixed sample. The application of Jensen-Shannon loss ensures that the augmented versions drift not too far from the original image, thereby enforcing consistency across all the training samples. We evaluate our method on various corruption benchmarks for robustness. The trained classifier nearly maintains calibration across different corruption severities. Overall, we observe that Dynamic Mixup performs one step ahead of the prior methods that performed remarkably well in improving the robustness of image classifiers. We foresee that fine-tuning Dynamic Mixup in the data processing pipeline will be beneficial in learning robust representations for transfer learning and settings where the models learn from unlabeled or synthetic data.

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

## A   JENSEN-SHANNON CONSISTENCY LOSS

Jensen–Shannon (JS) divergence is a symmetric measure of similarity between two probability distributions, based on Kullback–Leibler divergence (Kullback). The JS divergence is computed by first obtaining the average prediction probability:

$$M = (p_{org} + p_{dynmix1} + p_{dynmix2})/3, \tag{1}$$

where $p_{org} = \hat{p}(y|X_{org}), p_{dynmix1} = \hat{p}(y|X_{dynmix1}), p_{dynmix2} = \hat{p}(y|X_{dynmix2})$. Then, we compute the average KL divergence between each image version and the average prediction probability using:

$$JS(p_{org}; p_{dynmix1}; p_{dynmix2}) = \frac{1}{3}(KL[p_{org}\|M] + KL[p_{dynmix1}\|M] + KL[p_{dynmix2}\|M]). \tag{2}$$

The JS divergence measures the average information about the specifics of the distribution $(p_{org}, p_{dynmix1}, p_{dynmix2})$ from which a training image was sampled. We define total loss as Jensen-Shannon Consistency Loss, which is composed of the original Cross Entropy Loss ($\mathcal{L}$) for the classification task and the JS divergence for data augmentation:

$$\mathcal{L}(p_{org}, y) + \lambda JS(p_{org}; p_{dynmix1}; p_{dynmix2}). \tag{3}$$

## B   AUGMENTATIONS

We show the augmentations used for Dynamic Mixup in 5. The augmentations on CIFAR-10 and CIFAR-100 data consist of 11 types of operations: `flip(horizontal/vertical)`, `rotate`, `translate(X/Y)`, `shear(X/Y)`, `auto contrast`, `equalize`, `solarize`, and `posterize`. We apply an extra `downscale` operation while processing ImageNet data.

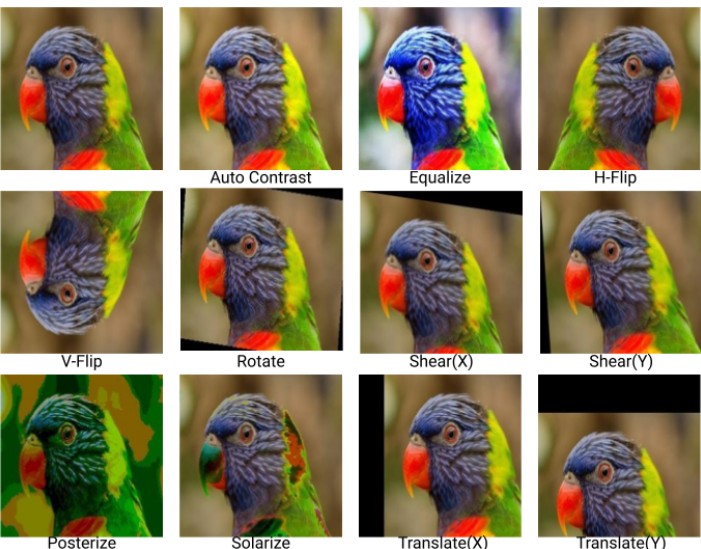

Figure 5: Augmentations used for Dynamic Mixup with 11 operations. We increased some operation intensities for illustration purposes.

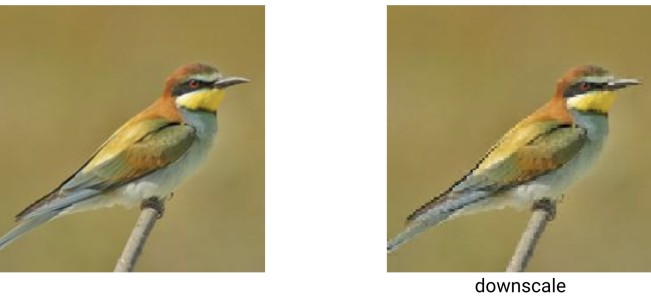

Figure 6: Sample image showing downscale operation on ImageNet.

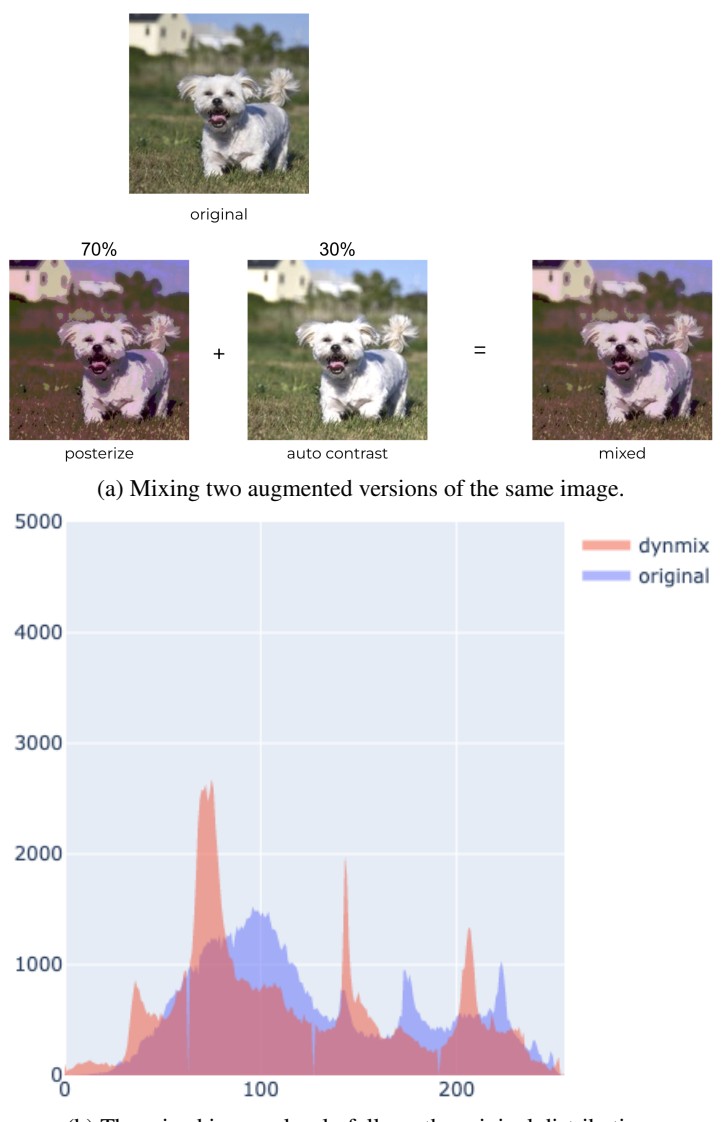

(a) Mixing two augmented versions of the same image.

(b) The mixed image closely follows the original distribution.

Figure 7: Dynamic Mixup approach with simple augmentation operations.

# C    MIXING APPROACH

We demonstrate our Dynamic Mixup approach with simple augmentation operations in Figure 7(a). Next, we show the results of mixing two distinct images in Figure 8(a). Previous methods have attempted to increase diversity by randomly mixing two disjunct images. We observe that the sample generated by Dynamic Mixup does not drift far from the original image upon mixing and closely captures the distribution shift (Figure 7(b). Meanwhile, even though we apply nominal augmentations, the resultant image exhibits a drift from the original image as visualized in Figure 8(b). Based on this, we infer that it can be challenging to interpret the results of mixing augmented versions of different images under heavy transformations. Sometimes it may not always be clear how the features of the two images have combined to produce the mixed image. Due to the unrealistic appearance of the generated images, the classifier is likely to make unreliable predictions. On the other hand, mixing augmentations of the same image not only allows learning diverse features under distribution shift but also results in images that are less prone to misclassification.

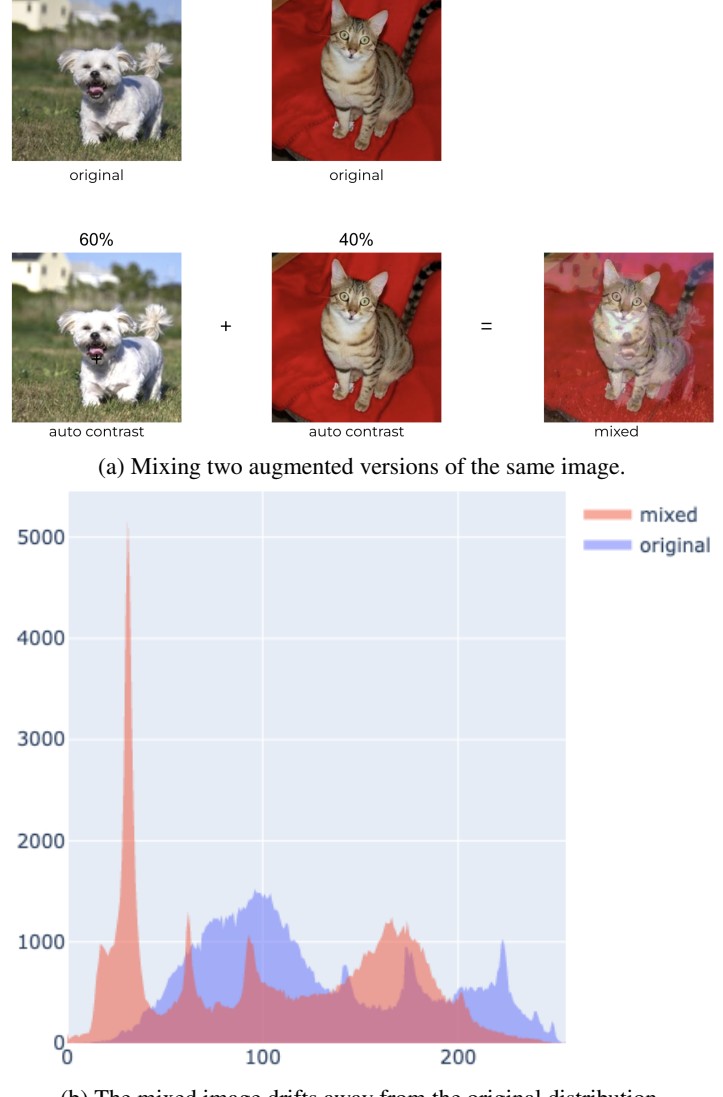

(a) Mixing two augmented versions of the same image.

(b) The mixed image drifts away from the original distribution.

Figure 8: A simple mixup approach with simple augmentation operations.

# D ADDITIONAL RESULTS

We include uncertainty estimates of the classifier's robustness on ImageNet using the RMS calibration error in Figure 9. Our method produces calibrated predictions to an extent across increasing levels of corruption severity as the classification error increases. Additionally, we provide the training loss results on CIFAR-100 using Wide ResNet and top-1 error rate in Figure 10. We see that the average classification error roughly approaches the clean error rate using Dynamic Mixup.

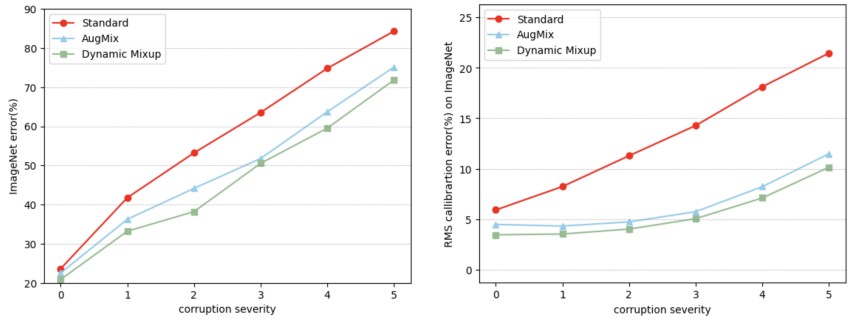

Figure 9: ImageNet error across severities, a severity level of zero corresponds to clean data.

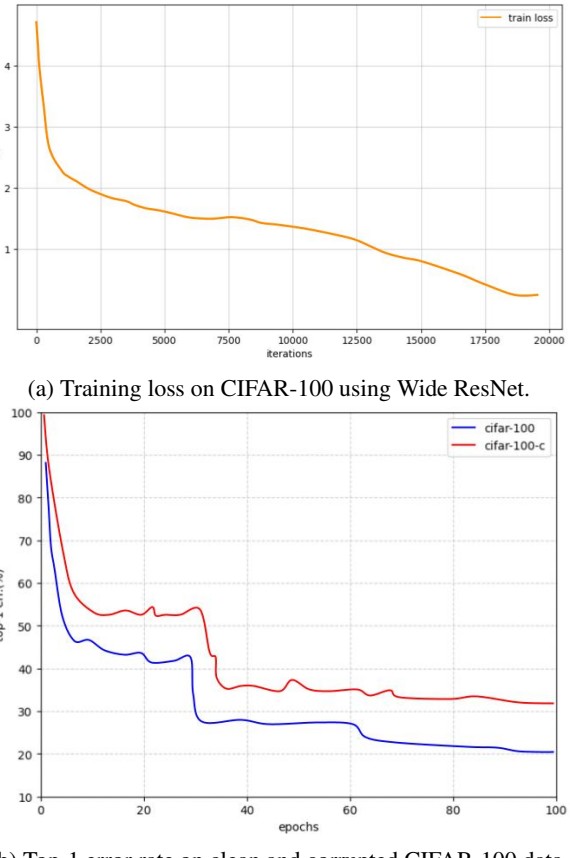

(a) Training loss on CIFAR-100 using Wide ResNet.

(b) Top-1 error rate on clean and corrupted CIFAR-100 data.

Figure 10: Average classification error using Dynamic Mixup approaches clean error rate.

