# OpenReview forum: "A Dynamic Mixup Approach Towards Improved Robustness of Classifiers"
_ICLR.cc/2024/Conference — ICLR 2024 Conference Withdrawn Submission_

### Official Review · Reviewer_g3Ch · 2023-10-24

**Soundness:** 3 good
**Presentation:** 3 good
**Contribution:** 1 poor
**Rating:** 3
**Confidence:** 4

**Summary:**

The paper introduces Dynamic Mixup to enhance the robustness of image classifiers. Unlike existing methods that rely on linear data augmentation techniques, Dynamic Mixup considers non-linearity in creating synthetic samples by dynamically mixing augmented images. This approach improves the model's ability to handle various real-world image corruptions and distribution shifts. Experimental results show that Dynamic Mixup outperforms previous methods in image and object classification tasks, significantly reducing corruption errors in datasets like CIFAR-10, CIFAR-100, ImageNet, and ImageNet-3DCC. The technique uses a simple data augmentation strategy with Jensen-Shannon loss to maintain consistency among training samples and achieve better robustness.

**Strengths:**

- This work covers an important topic, namely the robustness of neural networks against image corruption etc.
- The proposed Dynamic Mixup approach is straightforward and easy to understand.
- The paper shows sufficient ablation studies to justify its design choices.
- The proposed approach outperforms the compared methods.

**Weaknesses:**

- The reported numbers seem not to be consistent with previous works. When I compare the leaderboard in [A], the numbers for AugMix are not consistent with those reported in the paper. Why is that? Additionally, there are stronger methods that were not compared by the authors (DeepAugment+AugMix, ANT, etc.).
- The proposed method is only incremental to the previously proposed AugMix.
- These days the community uses more and more Vision Transformer architectures. How do the compared approaches perform for ViT architectures?

[A] https://github.com/hendrycks/robustness

**Questions:**

Please address the points in my weakness section.

---

> ### Author Response · Authors · 2023-11-16
>
> We are genuinely thankful and carefully consider the reviewer's comments to revise our manuscript. We would like to address a highlighted weakness in our response as we experiment with suggested methods.
>
> - The results provided in our paper align with the results provided in the original research paper [1].
>
> [1] AugMix: A Simple Data Processing Method to
> Improve Robustness and Uncertainty

---

### Official Review · Reviewer_XQmz · 2023-10-31

**Soundness:** 2 fair
**Presentation:** 3 good
**Contribution:** 2 fair
**Rating:** 3
**Confidence:** 4

**Summary:**

In this paper, authors propose a new mixing strategy, denoted as DynMix, which combines randomly augmented samples with linear coefficient and nonlinear interpolation. Thereby, improvements can be obtained compared to baselines.

**Strengths:**

The design of Dynmix is simple yet effective. Besides, Dynmix achieves gains in robustness and performs well in clean accuracy.

**Weaknesses:**

1. In essence, Dynmix still performs mixing by linear interpolation rather than nonlinear mixing and  ''dynamic mixing'' is not reflected in the method. Besides, the difference between MixUp and Dynmix lies in mixing clean samples with augmented samples and mixing different augmented samples. An important ablation study is missing in the paper, i.e., whether Dynmix outperforms the plain ''linear'' mixing strategy of (m * X_aug1 + (1-m) * X_aug2).

2. The comparison between Dynmix and multiple recent mixing-based baselines (e.g., PixMix[1], Puzzle Mix[2] and etc.) is missing, hence can not sufficiently prove the efficacy of Dynmix.

[1] Pixmix: Dreamlike pictures comprehensively improve safety measures, Hendrycks et al. CVPR 22.

[2] Puzzle Mix: Exploiting Saliency and Local Statistics for Optimal Mixup, Kim et al. ICML 20.

**Questions:**

1.  In Table 5, Dynmix applying JSD with single sample outperforms its counterparts in Table 1-2. Why not set Dynmix with single sample as the default setting?

---

> ### Author Response · Authors · 2023-11-16
>
> We thank the reviewer for their insightful comments and suggestions for improvement. As we take into consideration the provided research direction, we would like to answer the question raised as follows.
>
> - Although the results with a single sample do not significantly impact the performance, the motivation of this study is (a) to encourage diversity by considering the variability across different image samples (b) to learn richer representations (c) and work well in scenarios with limited data or less number of annotated data (semi-supervised learning).

---

### Official Review · Reviewer_Pd7H · 2023-10-31

**Soundness:** 3 good
**Presentation:** 2 fair
**Contribution:** 2 fair
**Rating:** 5
**Confidence:** 3

**Summary:**

The authors propose a new data augmentation technique called Dynamic Mixup. Dynamic Mixup is a data augmentation technique that enhances classification model robustness in unforeseen environments by applying diverse transformations and a mixing strategy, enabling the model to learn from various data distributions while maintaining consistency with the original image through Jensen-Shannon divergence.The authors claim robustness improvements and higher efficiency of Dynamic Mixup over previous augmentation techniques, and eval against baselines on ImageNet-C/P and some selected corruptions on CIFAR scale.

**Strengths:**

1. The paper is overall well written and easy to follow
2. The literature review is extensive and nicely done
3. There is some empirical evidence that the outlined method improves robustness especially on ImageNet-C/P and the other considered datasets.

**Weaknesses:**

The central claims of the paper, improved robustness and improved efficiency, seem a bit shaky under the proposed baseline choices.
1.   Need to include More recent Mixup Methods in Literature Review. And those the SOTA methods as baselines such as:

                a. AugMax: Adversarial Composition of Random Augmentations for Robust Training[Wang.et al 2021]

                b. Noisy Feature Mixup[Lim etal, 2021]
2.  It remains unclear whether the performance gains are significant, since no standard deviations of the performance metrics are provided.
3. The presentation of the proposed method's results may be misleading. Although the results show that the method tailors the model for specific types of corruption, the authors do not evaluate on adversarial attack such as PGD, Gaussian Noise.
	It is crucial to emphasize to the readers that the method does not necessarily make the model generalizable across diverse corruptions. The authors should address this concern and provide additional clarification in the paper.

Minor Weaknesses
	1. There is no evaluation on natural distribution shifts like ImageNet-R, ObjectNet, ImageNet-D, etc., although this would be quick to run and does not require re-training.

**Questions:**

1. Add more references regarding recent mixup methods in terms of robustness
2. Better to add a table for comparison between those mixup methods on CIFAR10/100, ImageNet  in terms of clean accuracy.
3. Can you please clarify "the residual term $m1m2(Xaug1 - Xaug2)$ captures non-linearity between the data points" How ? And what is the sample complexity is high, will such mixup policy require a huge computation burdern ?
4. Better to have more empirical evidence to testify the robustness of Dynamic Mixup such as perform PGD / Gaussian Noise to test the model.

---

> ### Author Response · Authors · 2023-11-16
>
> We are grateful for the reviewer's careful review of our work and their valuable feedback. As we work on the suggestions to improvise our work, we would like to address some of the comments here.
>
> - As per [1], “Other works create and capture the corruptions in the real world, e.g. ObjectNet. Although realistic, it requires significant manual effort and is not extendable. A more scalable approach is to use computer graphics-based 3D simulators to generate corrupted data [2] which can lead to generalization concerns. 3DCC aims to generate corruption as close to the real world as possible while staying scalable.” Hence, we tested our model on ImageNet-3DCC and provided the results in the paper.
>
> [1] 3D Common Corruptions and Data Augmentation
> [2] 3db: A framework for debugging computer vision models
>
> - The residual term m1m2(Xaug1-Xaug2) involves the multiplication of the weights m1​ and m2 by the difference between samples Xaug1 and Xaug2. This multiplicative interaction is a form of non-linearity because the effect of one weight depends on the value of the other, and their combined impact is not simply an additive or linear combination. When m1​ and m2​ are both non-zero, the interaction term introduces an additional degree of complexity to the model. The effect of this term is not constant; it varies based on the values of samples Xaug1 and Xaug2. This allows the model to capture more intricate patterns and relationships in the data that go beyond linear combinations.
>
> - We will add the results for PGD attacks, and the results on Gaussian Noise as mCE are provided in the paper.

---

### Official Review · Reviewer_UQyr · 2023-11-01

**Soundness:** 2 fair
**Presentation:** 2 fair
**Contribution:** 2 fair
**Rating:** 3
**Confidence:** 5

**Summary:**

The paper addresses the challenge of improving the robustness of image classifiers to data distribution shifts, which often leads to performance degradation. The authors introduce a novel data processing technique called "Dynamic Mixup," an adaptation of the mixup approach. Dynamic Mixup dynamically combines augmented image samples while considering the non-linearity between them. The idea is straightforward and easy to follow, the empirical scope of the method is too limited at this moment.

**Strengths:**

1. the main approach is very straightforward to follow.

2. the argument about addressing the non-linearity in data augmentation is very interesting, although the claim regarding existing method might not be true.

**Weaknesses:**

1. the main argument regarding existing method, such as "Current methods typically rely on data augmentation techniques to simulate distribution shifts based on image corruptions. These techniques mainly consider linearity to generate synthetic samples." might not be true.
     - relevant discussions have been made even in fairly aged papers [1] in terms of the linearity of mix-up, although looking at today, this method is unlikely to compete the methods that authors consider now.
     - the discussion also ignores large body of more recent papers that achieve SOTA performances on the relevant benchmarks, such as [2]

2. the main issue of this paper is the empirical scope, as the mixup paper, it did not compare more recent mix-up variants, such as [3], as a paper compares the SOTA leaderborder, it did not compare enough with the existing SOTA method on that line either, e.g., [2] and many others. The latest methods this paper compare is auto-aug and aug mix, both have couple years of history now.


[1]. Manifold Mixup: Better Representations by Interpolating Hidden States
[2]. Enhance the Visual Representation via Discrete Adversarial Training
[3]. Co-Mixup: Saliency Guided Joint Mixup with Supermodular Diversity

**Questions:**

I understand sometimes SOTA is not the only metric to evaluate a paper, but as a mix-up variant paper, it's quite necessary to show its SOTA competitiveness. I would encourage the authors to add more empirical results.

---

> ### Author Response · Authors · 2023-11-16
>
> We appreciate the reviewer's thoughtful comments and suggestions regarding our manuscript. We would like to address a few things mentioned as weaknesses as we continue to work on adding more empirical results as per the rest of the suggestions.
>
> - [1] leverages semantic linear interpolations. We have utilized the non-linear transfer characteristics of the input signal as in [2]. To the best of our knowledge, while linear interpolation techniques have been extensively studied, opportunities remain to include non-linear behavior during interpolation. Additionally, [1] uses a supervised learning paradigm where the loss term only uses binary cross-entropy and doesn’t focus on assessing how closely the augmented versions represent the training as well as the real-world data. In our work, we utilize the Jensen-Shannon Loss to enforce consistency among the mixed samples.
>
> [1] Manifold Mixup: Better Representations by Interpolating Hidden States
> [2] Chapter 3-Dynamic Range: Distortion and Noise